# Aggregate to Adapt: Node-Centric Aggregation for Multi-Source-Free Graph Domain Adaptation

## Abstract

Unsupervised graph domain adaptation (UGDA) focuses on transferring knowledge from labeled source graph to unlabeled target graph under domain discrepancies. Most existing UGDA methods are designed to adapt information from a single source domain, which cannot effectively exploit the complementary knowledge from multiple source domains. Furthermore, their assumptions that the labeled source graphs are accessible throughout the training procedure might not be practical due to privacy, regulation, and storage concerns. In this paper, we investigate multi-source-free unsupervised graph domain adaptation, i.e., exploring knowledge adaptation from multiple source domains to the unlabeled target domain without utilizing labeled source graphs but relying solely on source pre-trained models. Unlike previous multi-source domain adaptation approaches that aggregate predictions at model level, we introduce a novel model named GraphATA which conducts adaptation at node granularity. Specifically, we parameterize each node with its own graph convolutional matrix by automatically aggregating weight matrices from multiple source models according to its local context, thus realizing dynamic adaptation over graph structured data. We also demonstrate the capability of GraphATA to generalize to both model-centric and layer-centric methods. Comprehensive experiments on various public datasets show that our GraphATA can consistently surpass recent state-of-the-art baselines with different gains. Our source codes and datasets are available at https://anonymous.4open.science/r/GraphATA-C0D8.

## Keywords

Unsupervised Graph Domain Adaptation, Graph Neural Networks

**ACM Reference Format:**

Anonymous Author(s). 2025. Aggregate to Adapt: Node-Centric Aggregation for Multi-Source-Free Graph Domain Adaptation. In *Proceedings of the ACM Web Conference 2025 (WWW '25)*. ACM, New York, NY, USA, 13 pages. https://doi.org/XXXXXXX.XXXXXXX

## 1 Introduction

Web data is inherently complex, characterized by diverse entities and intricate relationships, making it challenging to mine meaningful insights. Graph algorithms play a pivotal role in numerous web applications, enabling more efficient representation [3, 49], analysis [5], and decision-making [13, 16], etc. While graph neural

networks (GNNs) [15, 20, 29, 46, 53] have achieved remarkable success across diverse tasks including node classification [20, 46, 49], traffic forecasting [60, 61], molecular property prediction [22, 42] and web-scale recommendation [11, 56], these GNN models exhibit substantial performance deterioration when applied to graphs with domain discrepancies [51]. To mitigate this gap and eliminate the need for label annotations, unsupervised graph domain adaptation [50, 51, 58] has been proposed to adapt the model by transferring knowledge from labeled source graph to unlabeled target graph. Existing graph domain adaptation approaches either learn domain invariant representations via adversarial training [40, 51] or explicitly minimize the domain distribution discrepancy [41, 50] to improve their generalization capability.

However, the above mentioned methods assume that the knowledge is specifically transferred from a single labeled source domain to an unlabeled target domain. Whereas, in the real world scenarios, data are often collected from multiple domains, which provides a range of complementary knowledge from different perspectives. This could significantly benefit target domains that do not strictly align with any single available source domain. For example, social networks might come from different countries and platforms with linguistic diversity. If the source networks are popular for a particular language like English or Spanish and the target network involves a mix of different languages, the adaptation can be tailored to target distribution by aggregating knowledge from multiple sources. To this end, Multi-Source Domain Adaptation (MSDA) [14, 37, 63] is introduced to learn from multiple source domains, allowing it to obtain complementary knowledge from various source domains and making it more resilient to domain shifts.

Unfortunately, recent MSDA approaches require labeled source data during the adaptation procedure, which might be impractical due to privacy as well as security concerns, especially when source data containing sensitive information, e.g., financial transactions [24] and medical diagnosis [9], etc. Therefore, it is imperative to investigate Multi-Source-Free Domain Adaptation (MSFDA) by relying solely on source pre-trained models without access to any labeled source data [1, 10, 39]. A simple yet straightforward solution for addressing MSFDA is to employ existing single-source-free domain adaptation methods [25, 27, 54] to adapt each source model individually, then the predictions from different source models are averaged to generate the final prediction. Nonetheless, it ignores the transferability of different source domains, since they may contribute differently to the target domain.

There are some recent studies that automatically assign weights to source predictions [1, 10, 39], where a larger value indicates higher transferability. Nevertheless, these aforementioned methods are designed for independent and identically distributed (i.e., iid) data, while the existence of non-iid graph-structured data poses great challenges to MSFDA that remain unexplored. In graphs, nodes are interconnected with each other through edges, forming

**Figure 1: A toy example, where GNN 1 excels in modeling shared interests, whereas GNN 2 is good at capturing geographical proximity. If node B has mixed connection types, simply combining the predictions from GNN 1 and GNN 2 is ineffective, as neither of the source pre-trained GNNs performs well in this scenario.**

complicated graph structure. Existing model-centric adaptation approaches, which learn a weight for each model, might not be adequate to capture the complementary semantics encoded by each model, leading to inaccurate combination of predictions. The main reason is that different nodes are associated with distinct local neighborhoods, thus globally aggregating source model predictions ignores the fine-grained node level disparity. For instance, source models are trained on two different social networks, e.g., one's connections emphasizing shared interests and the other one's links indicating geographical proximity. Then, we want to adapt these source models to classify node in target network, where neighboring connections might arise from shared interests as well as geographical proximity. As shown in Figure 1, the combination of model level predictions fails to adapt to different local patterns in the target network and results in sub-optimal performance. No matter how the predictions are merged, the outcome remains inaccurate because the individual predictions themselves are flawed. Thus, more devotion is required to effectively handle the graph domain adaptation task with fine-grained information.

To address the aforementioned key challenges, we propose a novel framework named GraphATA (**A**ggregate **T**o **A**dapt), *which performs node-centric adaptation through dynamically parameterizing each node with a unique graph convolutional matrix*. Instead of globally aggregating source model predictions, we conduct fine-grained adaptation by taking each node's local context information into consideration. At each layer, we generate a personalized graph convolutional matrix for each node by automatically aggregating source models' weight matrices based on its local neighborhood. Therefore, different nodes could have distinct optimal weight matrices, which is flexible to adapt to diverse patterns. Furthermore, sparse constraints are employed to filter out irrelevant information, since not all the source models are useful during the adaptation procedure. We have carried out extensive experiments including node as well as graph classification, and the experimental results demonstrate that our proposed GraphATA outperforms recent state-of-the-art baselines over widely used datasets.

In summary, the main contributions of this paper are as follows:

- To the best of our knowledge, we are the first to investigate the problem of multi-source-free unsupervised graph domain adaptation, which is a practical yet unexplored setting within the graph neural network community.

- We propose a node-centric adaptation framework that parameterizes each node with a personalized graph convolutional matrix according to its local context information, which enables a more generalizable model.

- Extensive experimental results show that GraphATA could achieve state-of-the-art performance across various public datasets with thorough ablation studies further validating the effectiveness of our node-centric adaptation.

## 2 Related Work

**Graph Neural Networks.** With the remarkable success in various graph related tasks, graph neural networks have drawn continuous attention in both academic and industrial communities. Different types of graph neural networks have been designed following the message passing paradigm, which can be categorized into spectral methods [2, 7, 20] and spatial methods [15, 46, 49]. Among them, GCN [20] performs convolution by approximating the Chebyshev polynomial [7] using its truncated first-order graph filter. GAT [46] utilizes an attention mechanism to learn different weights for dynamically aggregating node's neighborhood representations. GraphSAGE [15] introduces an inductive framework that generates representations by sampling and aggregating local representations. For more details, please refer to comprehensive surveys on graph neural networks [52, 65]. Despite their success, the performance of GNNs depends on high-quality labeled data, which can be challenging for graph-structured data. To address this issue, adapting models trained on label-rich source domains to unlabeled target domains has emerged as a promising solution.

**Unsupervised Domain Adaptation.** The goal of domain adaptation is to transfer knowledge from labeled source domains to unlabeled target domains. One key challenge lies in how to mitigate the domain shifts between source and target domains. To reduce the distribution discrepancy, most methods focus on learning domain invariant representations, which involve either explicit or implicit constraints. For example, some works [30, 59] employ maximum mean discrepancy or central moment discrepancy to explicitly minimize the distance between source and target distributions. Other studies [17, 31] utilize adversarial training to make the domain discriminator unable to differentiate source and target representations. Recently, there have been endeavors dedicated to unsupervised domain adaptation for non-iid graph-structured data. Particularly, UDAGCN [51] follows the adversarial training framework to learn domain invariant representations on graphs. GRADE [50] introduces the metric of graph subtree discrepancy to minimize the distribution shift between source and target graphs. SpecReg [58] designs spectral regularization for theory-grounded graph domain adaptation. Liu et al. [28] proposes an edge re-weighting strategy to reduce the conditional structure shift. Mao et al. [33] preserves target graph structural proximity and Zhang et al. [62] conducts collaborative adaptation in the scenario of single source-free graph domain adaptation. However, these methods cannot address the multi-source-free graph domain adaptation problem since they require labeled data or are unable to adapt complementary knowledge from multiple source domains.

**Multi-Source-Free Domain Adaptation.** MSFDA extends domain adaptation by transferring knowledge from multiple source

pre-trained models without accessing any source domain data. To capture the relationship among different source domains, various domain weighting strategies are utilized to estimate the contribution of each source domain to the target domain, including uniform weights, wasserstein distance-based weights and source domain accuracy-based weights [37, 47, 64, 66]. Due to the absence of source data, the above strategies are not applicable in the MSFDA setting. Towards this end, DECISION [1] and CAiDA [10] aggregate multiple source model predictions and construct pseudo labels for model adaptation. Shen et al. [39] propose to balance the bias-variance trade-off through domain aggregation, selective pseudo-labeling and joint feature alignment. Nonetheless, all these models are designed for independent and identically distributed data, which are not suitable for non-iid graph structured data. Moreover, aggregating model level predictions is insufficient to capture the highly diverse graph patterns, since the global weights cannot adequately reflect the importance of each node's local context. In contrast, our model performs adaptation at node granularity with aggregating weight matrices from multiple source models according to its local context.

## 3 Problem Statement

**Notations and Problem Definition.** In multi-source-free unsupervised graph domain adaptation, the goal is to jointly adapt multiple source pre-trained graph neural network models to a target graph without any labels. In this paper, we focus on adapting classification models with $K$ categories. Formally, let $\mathcal{G} = (\mathcal{V}, \mathcal{E}, \mathbf{X})$ denote the unlabeled target graph, where $\mathcal{V}$ and $\mathcal{E}$ are the node set and edge set respectively. $\mathbf{X} \in \mathbb{R}^{n \times d}$ indicates the node feature matrix, with $n$ representing the number of nodes and $d$ denoting the dimension of node features. Given a set of source pre-trained GNN models $\{\Phi_1, \Phi_2, \cdots, \Phi_m\}$, where the $i$-th model is trained using the graph from $i$-th source domain, we decompose each source model $\Phi_i$ into two basic components, i.e., the feature extractor $\phi_i : \mathcal{G} \rightarrow \mathbb{R}^{n \times d}$ encoding graph $\mathcal{G}$ into node representation space and the classifier $\psi_i : \mathbb{R}^{n \times d} \rightarrow \mathbb{R}^{n \times K}$ or $\mathbb{R}^K$ projecting node or graph representations into corresponding class labels. Hence, the source model $\Phi_i$ can be expressed as $\Phi_i = \phi_i \circ \psi_i$. Our ultimate problem can beformulated as follows:

*Given $m$ source trained graph neural network models $\{\Phi_1, \cdots, \Phi_m\}$ and an unlabeled graph $\mathcal{G}$ (node level task) or a set of unlabeled graphs $\{\mathcal{G}_1, \cdots, \mathcal{G}_n\}$ (graph level task) from target domain, our goal is to build a target model $\Phi_t$ that aggregates knowledge from multiple source models to achieve accurate predictions in target domain under distribution shifts.*

**Message Passing GNN Revisiting.** Most GNNs adopt the message passing framework [15, 20, 46], where each node iteratively aggregates representations from its local neighborhood. Specifically, the node $v$'s representation at layer $l$ can be calculated as follows:

$$\mathbf{h}_v^l = \sigma(\textsc{Agg}(\{\mathbf{h}_v^{l-1}\} \cup \{\mathbf{h}_u^{l-1}, \forall u \in \mathcal{N}(v)\}) \cdot \mathbf{W}^l), \quad (1)$$

where $\sigma(\cdot)$ is the activation function and $\textsc{Agg}(\cdot)$ represents the permutation-invariant aggregation function that aggregates message from its neighbors $\mathcal{N}(v)$. $\mathbf{W}^l$ denotes the convolutional matrix at layer $l$. The aggregation process in mainstream GNNs can be generalized as a weighted summation. For example, GCN [20] aggregates neighborhood representations using fixed weights inversely proportional to node degrees. GraphSAGE [15] utilizes a mean pooling aggregator, while GAT [46] employs an attention mechanism for learnable weighted aggregation. For graph classification task, we simply use global mean pooling and max pooling to assemble all the node representations in the graph. Advanced techniques like hierarchical graph pooling can also be utilized in this scenario [57].

## 4 The Proposed GraphATA Model

Figure 2 provides a comparison between existing model-centric methods and our proposed node-centric framework. *Specifically, model-centric adaptation approaches allocate a weight to each model, implying that all the nodes in the target graph share the same weight within each model.* Hence, it fails to reflect the unique characteristic of each individual target node, since the same model may exhibit varying capabilities when encoding different nodes. *In contrast, our node-centric adaptation framework GraphATA takes node disparity into consideration and parameterizes a unique convolutional matrix for each node to achieve fine-grained personalized adaptation.* Particularly, each node derives its own convolutional matrix by automatically aggregating matrices from multiple source GNN models based on its local neighborhood, which results in more generalizable model. Subsequently, we will elaborate the details of the proposed modules.

**Node Neighborhood Disparity.** We start by investigating the local context of each node within the graphs. Different nodes typically exhibit diverse structural patterns as they are not uniformly distributed across the graph. To characterize this property, we conduct a thorough examination of the node's homophilic and heterophilic patterns through the lens of node homophily ratio, which is a widely adopted metric that quantifies the proportion of a node's neighbors having the same class label [23, 32, 36]. It is formally defined as follows:

$$h_v = \frac{|\{u \in \mathcal{N}(v) : y_u = y_v\}|}{|\mathcal{N}(v)|}, \quad (2)$$

where $\mathcal{N}(v)$ represents node $v$'s neighbors set and $y_v$ indicates the class label for node $v$. Figure 3 demonstrates the node homophily ratio distributions on three social graphs from Twitch datasets (Section 5.1). We can observe that *(1) all three graphs manifest a mixture of homophilic as well as heterophilic patterns; (2) the patterns' distributions vary significantly across different graphs.* Thus, existing model-centric methods [1, 10] overlook each node's neighborhood disparity and the allocated weights might be sub-optimal. The above observations motivate us to perform fine-grained node-centric adaptation.

**Node-Centric Adaptation.** In the above investigation, we recognize the necessity of adapting to the local context of each individual node. To achieve this goal, we propose to assign distinct matrices to different nodes by aggregating convolutional matrices from the source pre-trained models, rather than aggregating model predictions. Specifically, different pre-trained models in the source domains have encapsulated different semantic information, which demonstrate varying capabilities in encoding the local context of each target node. For each node $v$, we utilize a straightforward yet effective way to represent its local contextual information at layer

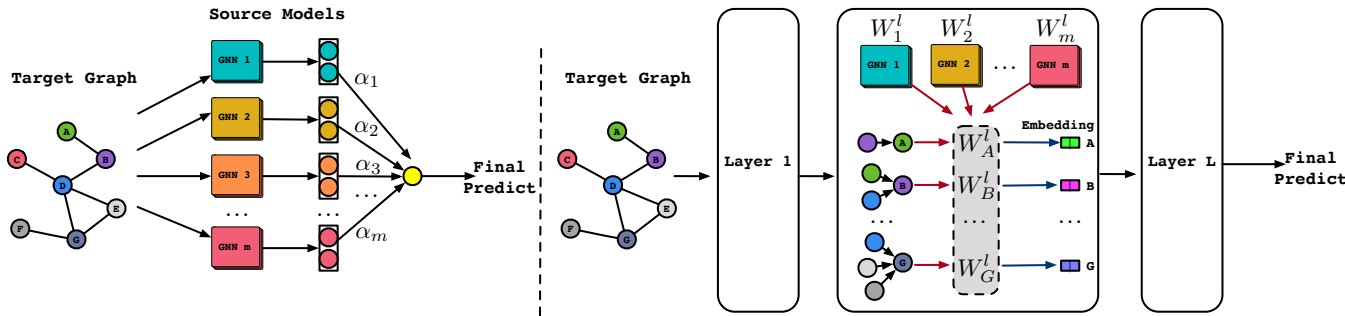

(a)Existing Model Centric Adaptation

(b) GraphATA: Node Centric Adaptation

**Figure 2: An illustrative comparison between existing model-centric methods and our proposed node-centric framework. (a) The target prediction is the weighted combination of source models' predictions. (b) GraphATA performs fine-grained adaptation by considering each node's unique characteristic. The grey box with dash lines shows the personalized convolutional matrix for each node at layer $l$.**

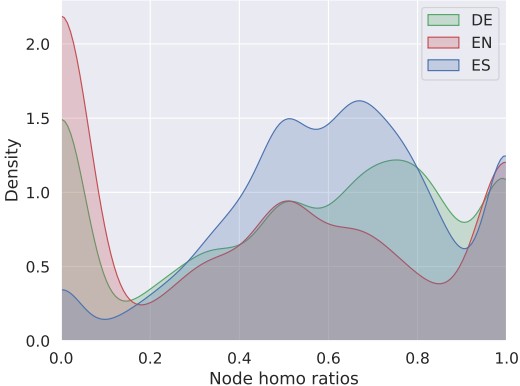

**Figure 3: Node homophily ratio distributions.**

$l$ as follows:

$$\mathbf{c}_v^l = \text{MEAN}(\{\mathbf{h}_u^{l-1}, \forall u \in \mathcal{N}(v)\}), \quad (3)$$

where we adopt mean operation to pool its neighbor's representation $\mathbf{h}_u^{l-1}$ from previous layer and $\mathbf{c}_v^l \in \mathbb{R}^{d_{l-1}}$. More alternative options are presented at Appendix E.

After having obtained the local context $\mathbf{c}_v^l$, we generate a personalized graph convolutional matrix for node $v$ as follows:

$$\mathbf{W}_v^l = \sum_{i=1}^m \alpha_{vi}^l \Lambda(\mathbf{c}_v^l) \mathbf{W}_i^l + \lambda \mathbf{W}_g^l, \quad (4)$$

where $\mathbf{W}_i^l \in \mathbb{R}^{d_{l-1} \times d_l}$ represents the convolutional matrix from the $i$-th source pre-trained GNN model at layer $l$ and $\Lambda(\mathbf{c}_v^l)$ is the $d_{l-1} \times d_{l-1}$ diagonal matrix with its elements setting as $\mathbf{c}_v^l$. The attentive coefficient $\alpha_{vi}$ characterizes the importance of each source domain model when adapting to node $v$. We further incorporate a global parameter $\mathbf{W}_g^l$ shared by all the nodes in the $l$-th layer to capture the global general patterns and $\lambda$ is a trade-off parameter. Therefore, our derived personalized $\mathbf{W}_v^l$ considers not only local but also global aspects of the graph, making it more adaptable to different types of distribution shifts.

**Sparse Attention Selection.** In Equation (4), although $\mathbf{W}_v^l$ automatically aggregates the convolutional matrices from multiple source models according to its local context, we posit that not all the source domain models are useful, which is known as "negative transfer" [4, 48]. To combat this issue, we aim to filter out detrimental models and preserve a sparse mixture of effective models via attention coefficients. Particularly, we utilize a shared linear transformation parametrized by $\mathbf{a}^l \in \mathbb{R}^{d_l}$ to quantify the trustworthy and reliability of each model when adapting to the target node's local contextual information $\mathbf{c}_v^l$. At each layer, the attention score can be calculated as follows:

$$\alpha_{vi} = \text{Attention}(\mathbf{a}, \mathbf{c}_v, \mathbf{W}_i) = \mathbf{a}^\top(\mathbf{W}_i^\top \mathbf{c}_v), \quad (5)$$

where the superscripts are omitted for simplicity. Additionally, we normalize the scores to ensure that $\alpha_{vi} \in [0, 1]$ and $\sum_{i=1}^m \alpha_{vi} = 1$. One commonly utilized approach is to employ the softmax function; however, it always produces non-zero values, which fails to yield the desired selective results.

Inspired by recent successes on sparse activation functions, we choose to adopt the sparsemax function [34], which preserves the crucial properties of the softmax function and generates sparse distributions. It projects the input onto the probability simplex as follows:

$$\text{sparsemax}(\boldsymbol{\alpha}) = \underset{\mathbf{x} \in \Delta^{m-1}}{\arg\min} \|\mathbf{x} - \boldsymbol{\alpha}\|^2, \quad (6)$$

where the simplex $\Delta^{m-1} = \{\mathbf{x} \in \mathbb{R}^m | \mathbf{1}^\top \mathbf{x} = 1, \mathbf{x} \geq \mathbf{0}\}$. Its closed-form solution can be formulated as follows:

$$\text{sprasemax}_i(\boldsymbol{\alpha}) = [\alpha_i - \tau(\boldsymbol{\alpha})]_+, \quad (7)$$

where $[x]_+ = \max\{0, x\}$ and $\tau(\cdot)$ is the threshold function that satisfies $\sum_j [\alpha_j - \tau(\boldsymbol{\alpha})]_+ = 1$. To compute $\tau(\boldsymbol{\alpha})$, we first sort $\boldsymbol{\alpha}$ in descending order: $\alpha_1 \geq \alpha_2 \geq \cdots \geq \alpha_m$, and define $\eta = \max\{1 \leq j \leq m | \alpha_j > \frac{1}{j}(\sum_{i=1}^j \alpha_i - 1)\}$. Then, we have $\tau(\boldsymbol{\alpha}) = \frac{\sum_{i=1}^\eta \alpha_i - 1}{\eta}$. The sprasemax function truncates the values below the threshold to zero and shifts the remaining values by this threshold. Detailed proof is provided in Appendix A.

**Model Optimization.** To optimize the model's parameters, we leverage predictions from the nearest neighbors to generate pseudo labels. For the stability of the learning procedure, we maintain

a target representation bank $\mathcal{R} = [\tilde{\mathbf{h}}_1, \cdots, \tilde{\mathbf{h}}_n]$ and a prediction bank $\mathcal{P} = [\tilde{\mathbf{p}}_1, \cdots, \tilde{\mathbf{p}}_n]$ through a momentum updating manner as follows:

$$\tilde{\mathbf{h}}_i = (1 - \gamma)\tilde{\mathbf{h}}_i + \gamma \mathbf{h}_i, \ \tilde{\mathbf{p}}_i = (1 - \gamma)\tilde{\mathbf{p}}_i + \gamma \mathbf{p}_i, \tag{8}$$

where $\gamma$ denotes the smoothing parameter setting as 0.9 by default. $\mathbf{h}_i \in \mathbb{R}^d$ and $\mathbf{p}_i \in \mathbb{R}^K$ are the outputs of feature extractor $\phi_t$ and classifier $\psi_t$, respectively. Then, for each target representation $\mathbf{h}_i$, we extract $r$ nearest neighbors from representation memory bank $\mathcal{R}$ according to their cosine similarities. With the nearest neighborhood information, the pseudo label distribution of sample $i$ can be obtained by aggregating the predicted class distributions of these nearest neighbors in memory bank $\mathcal{P}$ as follows:

$$\hat{\mathbf{y}}_i = \mathbb{1}\left[\arg\max_k \left(\frac{1}{|\mathcal{S}(i)|} \sum_{j \in \mathcal{S}(i)} \tilde{\mathbf{p}}_j\right)\right], \tag{9}$$

where $\mathbb{1}[\cdot]$ represents the one-hot transformation function and $\mathcal{S}(i)$ is a set of $r$ nearest neighbors' indices for sample $i$. Thus, we could update the model's parameters by minimizing the cross entropy loss between the generated pseudo labels and the predicted class distributions as follows:

$$\mathcal{L}_{cls} = -\frac{1}{n}\sum_{i=1}^{n}\sum_{k=1}^{K}\hat{\mathbf{y}}_{i,k}\log(\mathbf{p}_{i,k}). \tag{10}$$

Additionally, we further encourage the prediction to be individually certain and globally diverse [25] to avoid the degenerated prediction. Therefore, we minimize the entropy for each individual sample while maximizing the entropy for each class, which is expressed as follows:

$$\mathcal{L}_{reg} = \left[\frac{1}{n}\sum_{i=1}^{n}\mathcal{H}(\mathbf{p}_i)\right] - \mathcal{H}\left(\frac{1}{n}\sum_{i=1}^{n}\mathbf{p}_i\right). \tag{11}$$

Among them, $\mathcal{H}(\mathbf{p}_i) = -\sum_{k=1}^{K}\mathbf{p}_{i,k}\log(\mathbf{p}_{i,k})$ denotes the entropy function. Finally, we can obtain the overall objective function as follows and the training procedure is presented in Algorithm 1:

$$\mathcal{L} = \mathcal{L}_{cls} + \mathcal{L}_{reg}. \tag{12}$$

**Model Analysis.** We discuss how our GraphATA generalizes to existing state-of-the-art methods. *(1) Relation with layer-centric approaches.* In particular, when setting $\mathbf{c}_v = \mathbf{1}$ and $\mathbf{W}_g = \mathbf{0}$, we suppress the awareness of each node's local contextual information, thus every node will have the same matrix expressed as $\mathbf{W}_{1,\cdots,n}^l = \sum_{i=1}^{m}\alpha_i\mathbf{W}_i^l$ within each layer. It essentially performs a weighted combination of the node representations propagated at each layer. Taking GCN [20] as an example, we have $\mathbf{H}^l = \sigma(\tilde{\mathbf{A}}\mathbf{H}^{l-1}\sum_{i=1}^{m}\alpha_i\mathbf{W}_i^l) = \sum_{i=1}^{m}\alpha_i\sigma(\tilde{\mathbf{A}}\mathbf{H}^{l-1}\mathbf{W}_i^l)$, where $\sigma(\cdot)$ is the ReLU activation function and $\tilde{\mathbf{A}}$ indicates the normalized adjacent matrix with self-loops. *(2) Relation with model-centric methods.* If we further restrict the information aggregation to the last layer $L$, i.e., the allocated weights are only employed for aggregating the predictions from each model and there is no information fusion in the intermediate layers, the simplified GraphATA degenerates to model-centric methods. *In summary, the design of GraphATA enjoys various benefits by taking each node's local context into consideration, and the current layer-centric as well as model-centric approaches are its special cases.*

**Table 1: Dataset Statistics.**

| Datasets | #Nodes | #Edges | #Feat | #Class |
|---|---|---|---|---|
| CSBM | 8,000~8,000 | 607,699~752,776 | 128 | 4 |
| Twitch | 1,912~9,498 | 31,299~153,138 | 3,170 | 2 |
| Citation | 5,484~9,360 | 8,117~15,556 | 6,775 | 5 |
| Proteins | ~39.06 | ~72.82 | 4 | 2 |
| Mutagenicity | ~30.32 | ~30.77 | 14 | 2 |
| Frankenstein | ~16.90 | ~17.88 | 780 | 2 |

**Complexity Analysis.** Suppose that we have a graph with $n$ nodes and $e$ edges, the node representation dimension is set as $d$ and the graph neural network has $L$ layers. Calculating the local contextual information in each layer has the cost of $O(ed)$. Then, if we have $m$ source models from different domains, the time complexity of generating sparse attention weights for each node is $O(md + m\log(m))$. Over $L$ layers, the feature encoder has the time complexity of $O(Lnd^2 + Led)$. Computing pseudo labels involves obtaining nearest neighbors, which takes $O(dn\log(n))$ using k-d tree. Thus, the overall time complexity of our model falls within the same range as that of vanilla graph neural network. Detailed comparisons with baselines are presented at Appendix D.

## 5 Experiments

### 5.1 Datasets

To fully validate the effectiveness of our proposed GraphATA, we perform node and graph classification tasks from various domains. The summary of dataset statistics is presented in Table 1 and the details are described as follows:

*CSBM* is a synthetic dataset, which is composed of four graphs generated by a 4-class contextual stochastic block model [8]. Each class contains 2,000 nodes in each graph. To synthesize different conditional structural shift, we fix the intra-class edge probability $p$ and vary the inter-class probability $q$ to generate different graphs. The node attributes are sampled from multivariate normal distributions with different mean vectors. The detailed process is described in Appendix B.

*Twitch*[1] [38] consists of six social networks from different regions, i.e., Germany (DE), England (EN), Spain (ES), France (F), Portugal (P) and Russia (R). The nodes represent users, while the edges denote their friendships. We construct node attributes from various factors such as users' gaming activities, preferences, geographic locations and streaming habits, etc. All the nodes are classified into two categories based on whether they use explicit language.

*Citation* [43] contains three research paper citation networks from different platforms and time periods. Particularly, DBLPv7 (D) is extracted from the DBLP database spanning the years 2004 to 2008; ACMv9 (A) comprises papers from ACM database between years 2000 and 2010; Citationv1 (C) is derived from MAG database prior to the year 2008. We categorize each paper into one of the five classes, i.e., DB, AI, CV, IS and Networking.

---

[1]https://snap.stanford.edu/data/twitch-social-networks.html

For graph classification task, we utilize three widely adopted datasets from TUdatasets[2] [35], i.e., **Proteins**, **Mutagenicity** and **Frankenstein**. To differentiate the distribution shifts in the datasets, we partition each dataset into four disjoint groups based on their density. Specifically, we first sort all the graphs in each dataset by their density in an ascending order, and then divide them into four equally disjoint groups.

## 5.2 Baselines

We compare our proposed GraphATA with four groups of baselines including *source-needed*, *no-adaptation*, *single-source-free* and *multi-source-free* domain adaptation methods. The detailed introduction is elaborated as follows:

**Source-needed.** Approaches in this category leverage the labeled source domains' samples to explicitly address the distribution shifts. We consider two multi-source-needed models MDAN [63], $M^3$SDA [37] and three single-source-needed models UDAGCN [51], GRADE [50] and SpecReg [58] as our baselines. For single-source-needed methods, we merge all the samples from different source domains into a single unified source domain.

**No-adaptation.** This group of baselines include widely used graph neural network models like GCN [20], GraphSAGE [15], GAT [46] and GIN [53]. The model is trained on each labeled source domain and then directly evaluated on the target domain. We output final predictions by taking an average of soft predictions from all the source models.

**Single-source-free.** We also extend existing single-source-free models including SHOT [25], BNM [6], ATDOC [26], NRC [54], JMDS [21], GTrans [18], SOGA [33], GraphCTA [62] and TPDS [44] to work in the scenario of multi-source-free domain adaptation. To achieve this goal, we utilize ensemble averaging to integrate soft predictions from all adapted source models.

**Multi-source-free.** Methods in this classes are recent state-of-the-art multi-source-free domain adaptation baselines such as DECISION [1], CAiDA [10] and MSFDA [39]. They automatically assign suitable weights to each model's predictions for final predictions. These models are originally designed for i.i.d images and we replace its backbone to adapt them for graph structured data.

**Experimental Settings.** Following recent works [51, 55], we randomly partition the samples in each source domain into training set (80%), validation set (10%) and test set (10%), respectively. The source model is first trained using the training set and its hyper-parameters are fine-tuned on the validation set. Then, we conduct sanity check on the test set to ensure that it is well-trained on the labeled source domain. The final performance is evaluated on the entire target domain. For baselines, we employ the source codes released by the authors and utilize the same graph neural network backbone with identical layers. The node representation dimension is set as 128 for node classification and 64 for graph classification tasks. We implement our proposed GraphATA with Pytorch Geometric [12] and the parameters are optimized with Adam [19]. The optimal learning rate and weight decay are searched in the set of $\{0.1, 0.01, 0.001, 1e^{-4}\}$. We keep the smoothing parameter $\gamma$ for

---

[2]https://chrsmrrs.github.io/datasets/docs/datasets/

memory banks as 0.9 by default and search the trader-off hyper-parameter $\lambda$ within the range of [0, 1]. More implementation details are given at Appendix C.

## 5.3 Results and Analyses

We show the results of node classification and graph classification in Table 2. Additional experiments on large-scale datasets are presented in Table 8 and Table 9 in Appendix E. All the experiments are repeated 5 times and we report the mean accuracy with standard deviation. In summary, we have the following key observations.

First, our proposed GraphATA surpasses all the baselines across various adaptation tasks with different margins. For instance, we achieve 12.20% average relative gains in the scenario of A,D→C compared with the naive no-adaptation method GCN. This implies that simply taking an average of the source model predictions cannot obtain satisfied performance, which is because different domains might contribute differently to the target domain. Therefore, it is important to aggregate information from multiple source domains with suitable weights. We also notice that GAT exhibits relatively poorer performance within this group. The reason can be attributed to the distribution shifts between source and target domains, as the optimal attention weights in source domains become less suitable for the target domain.

Second, when further compared with source-need methods that utilize labeled source data during the training process, our model can still outperform them by significant margins. Meanwhile, single-source-needed baselines, which consolidate all the samples into one large source domain, occasionally beat approaches specifically designed for multi-source domain adaptation like MDAN and $M^3$SDA in several settings. It justifies the necessity of aggregating rich information from multiple source domains, since they may contain complementary information for the target domain.

Third, all the multi-source extensions of single-source-free models demonstrate superior performance compared with non-adapted graph neural networks, even though they utilize the same ensemble strategy. These results suggest that domain adaptation is a promising way to mitigate the distribution shifts across different domains. However, there does not exist a clear winner that consistently outperforms the others within this category, because taking average of the predictions might be sub-optimal in some scenarios.

Finally, our GraphATA consistently exceeds the strongest baselines tailored for multi-source-free domain adaptation, such as DECISION, CAiDA and MSFDA. This stems from the fact that our model conducts fine-grained adaptation by comprehensively capturing each node's local context information, while existing model-centric approaches only aggregate information at the prediction level and overlook the fine-grained information. In challenging datasets with significant domain shifts, such as the Frankenstein and Protein datasets, traditional multi-source-free models struggle due to the likelihood of negative transfer. In contrast, our proposed GraphATA could filter out irrelevant models and reduce the impact of negative transfer, leading to more precise adaptation.

## 5.4 Ablation Studies

**The Effect of Different Modules.** To fully investigate the contribution of each component in our proposed GraphATA model,

**Table 2: Average node and graph classification performance in terms of accuracy with standard deviation (%). OOM means out-of-memory. Some of the models are specifically designed for node classification task; therefore, we denote them with '-' for graph classification task.**

| | Node Classification | | | | | | Graph Classification | | |
| | CSBM | Twitch | | Citation | | | Proteins | Mutagenicity | Frankenstein |
| Models | C1,C2,C3→C4 | R,P,F,ES→DE | R,P,F,ES→EN | A,C→D | C,D → A | A,D → C | P1,P2,P3→P4 | M1,M2,M3→M4 | F1,F2,F3→F4 |
|---|---|---|---|---|---|---|---|---|---|
| MDAN [63] | 76.49±0.39 | 61.78±0.33 | 49.52±0.35 | 70.38±0.17 | 65.31±0.09 | 73.92±0.13 | 48.56±1.34 | 63.85±3.11 | 49.02±1.28 |
| M$^3$SDA [37] | OOM | 59.75±0.64 | 53.71±0.65 | 70.22±0.16 | 64.71±0.33 | 71.30±0.16 | 52.15±1.28 | 69.24±1.46 | 48.48±3.67 |
| UDAGCN [51] | 72.18±0.12 | 43.34±0.30 | 48.88±0.22 | 70.54±0.04 | 61.67±0.01 | 69.72±0.03 | - | - | - |
| GRADE [50] | 78.12±0.21 | 53.52±0.07 | 46.80±0.04 | 74.37±0.49 | 70.92±0.04 | 79.48±0.05 | - | - | - |
| SpecReg [58] | 79.14±0.38 | 42.85±2.72 | 47.21±1.34 | 76.42±0.66 | 72.69±0.51 | 78.67±2.46 | - | - | - |
| GCN [20] | 73.18±0.45 | 53.57±0.49 | 47.76±0.11 | 70.92±0.15 | 64.64±0.12 | 72.98±0.18 | 47.48±2.01 | 68.09±1.44 | 46.40±1.76 |
| SAGE [15] | 76.91±0.78 | 42.24±0.43 | 45.89±0.15 | 68.92±0.28 | 61.25±0.38 | 67.77±0.17 | 51.65±1.44 | 66.78±1.69 | 49.24±0.42 |
| GAT [46] | 71.59±0.36 | 39.56±0.32 | 45.44±0.93 | 63.85±1.08 | 56.12±1.51 | 62.73±1.09 | 46.04±1.87 | 65.33±2.36 | 47.25±1.83 |
| GIN [53] | 76.92±0.25 | 40.75±0.61 | 45.43±0.26 | 67.02±0.36 | 59.50±0.25 | 70.34±0.23 | 40.79±4.67 | 67.42±1.69 | 46.07±1.91 |
| SHOT [25] | 83.82±0.25 | 64.01±0.09 | 57.97±0.07 | 75.34±0.13 | 68.71±0.41 | 77.92±0.05 | 49.56±2.33 | 60.36±1.67 | 48.45±1.38 |
| BNM [6] | 81.79±0.47 | 64.11±0.23 | 57.79±0.20 | 75.17±0.16 | 68.81±0.12 | 77.38±0.11 | 49.13±6.02 | 60.18±2.28 | 48.80±0.68 |
| ATDOC [26] | 64.72±0.33 | 57.92±0.97 | 50.94±2.80 | 72.67±0.63 | 65.52±1.16 | 76.12±0.29 | 52.01±2.28 | 56.75±4.21 | 45.49±3.30 |
| NRC [54] | 84.42±0.16 | 63.76±0.09 | 57.66±0.05 | 73.13±0.08 | 69.52±0.25 | 77.51±0.22 | 52.30±3.56 | 60.55±1.15 | 47.60±1.06 |
| JMDS [21] | 63.74±0.39 | 64.00±0.05 | 46.78±0.17 | 72.21±0.37 | 65.26±0.18 | 74.34±0.29 | 49.74±2.00 | 64.61±0.30 | 51.82±5.57 |
| GTrans [18] | 64.81±0.89 | 62.54±0.02 | 57.29±0.07 | 73.67±0.65 | 69.59±2.06 | 78.74±0.43 | - | - | - |
| SOGA [33] | 65.50±0.55 | 58.24±0.85 | 46.98±0.48 | 73.38±0.17 | 66.96±0.33 | 78.06±0.23 | - | - | - |
| GraphCTA [62] | 83.21±0.43 | 62.70±0.24 | 56.28±0.16 | 75.53±0.30 | 70.91±0.63 | 79.59±0.22 | - | - | - |
| TPDS [44] | 84.03±0.23 | 63.14±0.10 | 57.83±0.14 | 71.63±0.12 | 68.45±0.23 | 75.87±0.21 | - | - | - |
| DECISION [1] | 88.02±0.28 | 63.92±0.26 | 57.92±0.12 | 76.02±0.67 | 70.40±0.31 | 79.39±0.35 | 50.21±0.74 | 57.89±2.16 | 48.50±1.87 |
| CAiDA [10] | 87.96±0.94 | 63.73±0.49 | 57.73±0.45 | 75.75±0.42 | 70.25±0.54 | 79.70±0.74 | 50.93±3.08 | 53.74±5.27 | 46.25±2.66 |
| MSFDA [39] | 88.94±0.62 | 54.42±5.50 | 55.77±0.58 | 76.03±1.27 | 68.60±2.94 | 78.63±0.92 | 49.64±1.62 | 62.21±4.00 | 50.59±1.32 |
| **GraphATA** | **90.14±0.36** | **66.71±0.39** | **59.56±0.14** | **78.45±0.87** | **73.17±0.52** | **82.33±0.75** | **56.47±1.48** | **71.26±2.33** | **54.15±1.29** |

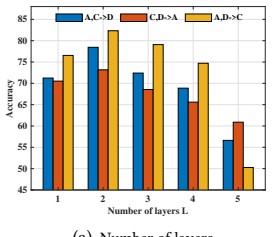

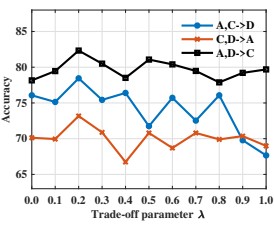

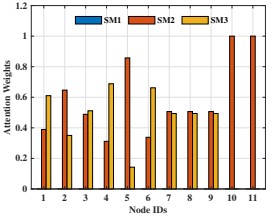

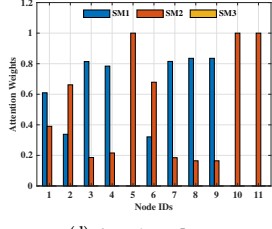

(a) Number of layers  (b) Trade-off param $\lambda$  (c) Attention at Layer 1  (d) Attention at Layer 2

**Figure 4: Hyper-parameter sensitivity analysis and attention weights visualization.**

**Table 3: GraphATA results with different components.**

| Models | A,C→D | C,D→A | A,D→C |
|---|---|---|---|
| GraphATA | 78.45±0.87 | 73.17±0.52 | 82.33±0.75 |
| GraphATA$_{\text{softmax}}$ | 73.99±1.61 | 71.13±0.85 | 80.50±0.50 |
| GraphATA$_{\text{MC}}$ | 74.97±0.65 | 70.23±0.69 | 79.70±0.81 |
| GraphATA$_{\text{LC}}$ | 74.60±0.97 | 68.91±0.58 | 77.38±0.61 |

we conduct a series of ablation studies on citation datasets. We first show the rationality and effectiveness of utilizing sparse attention to selectively aggregate convolutional matrices from multiple source pre-trained models. When replacing the sparsemax function with softmax function in Eq. (6) (termed as GraphATA$_{\text{softmax}}$), its performance degrades 4.46% and 2.04% in Table 3, respectively. This indicates that not all of the source models are useful, and filtering out irrelevant information would be beneficial. We further degenerate Eq. (4) to model-centric method (restricting the information aggregation to the last layer) and layer-centric method (setting $\mathbf{c}_v = 1$, $\mathbf{W}_g = \mathbf{0}$), which are denoted as GraphATA$_{\text{MC}}$ and GraphATA$_{\text{LC}}$. When compared with GraphATA, their performance decreases about 2.93% ∼ 4.26%. This justifies the advantages of conducting fine-grained node-centric adaptation. More ablation studies can be found at Appendix E.

**Hyperparameter Analysis and Attention Visualization.** We also show the impacts of several key hyper-parameters in Figure 4(a) and Figure 4(b). Particularly, when setting number of layers $L = 2$ and $\lambda = 0.2$, our model could always obtain the satisfied performance. To demonstrate the uniqueness of each node's graph convolutional matrices, we randomly sample a graph with 11 nodes and 11

**Table 4: GraphATA results with different loss functions.**

| Models | A,C→D | C,D→A | A,D→C |
|---|---|---|---|
| GraphATA | 78.45±0.87 | 73.17±0.52 | 82.33±0.75 |
| GraphATA$_{\text{w/o}\,\mathcal{L}_{\text{reg}}}$ | 74.37±1.20 | 69.30±1.52 | 80.02±0.45 |
| GraphATA+SHOT | 74.78±1.07 | 69.77±0.44 | 78.03±0.19 |
| GraphATA+BNM | 75.90±0.18 | 69.09±0.12 | 77.77±0.40 |
| GraphATA+CAiDA | 73.70±0.29 | 68.90±0.27 | 76.98±0.98 |

edges from the Mutagenicity graph classification dataset. The two-layer target model is optimized in the setting of M1,M2,M3→M4. Then, we plot the attention weights for each node at each layer in Figure 4(c) and Figure 4(d). As we can see, when aggregating convolutional matrices from source domains, distinct nodes have different inclinations both within each layer and across different layers. For instance, our model mainly aggregates information from SM2 and SM3 at layer 1, while it integrates information from SM1 and SM2 at layer 2. Both two layers do not use all of the three pre-trained models. We also note that node 10 and node 11 consistently utilize SM2's convolutional matrix across both layers. In contrast, the remaining nodes assign different weights to SM2 in layer 2 compared with layer 1. This highlights different source models indeed play distinct roles in modeling different nodes in the graph, therefore it is necessary to take fine-grained node-wise adaptation into consideration to effectively address the distribution shifts problem. By tailoring the adaptation to the unique characteristics of each node, we can more accurately align with the target graph's local structures, leading to more robust generalization.

**The Effect of Different Loss Functions and GNN Architectures.** Our GraphATA can also seamlessly integrate with various loss functions. To show the effectiveness of our proposed neighborhood consistency loss function, we replace it with three widely adopted loss functions from existing domain adaptation methods including SHOT [25], BNM [6] and CAiDA [10]. Specifically, SHOT employs self-clustering to generate pseudo labels, which overlooks its local neighborhood information. BNM adopts nuclear norm maximization to improve the model's discriminability and diversity, while CAiDA utilizes various strategies to select confident anchors and then construct pseudo labels through them. The results are shown in Table 4 and we can conclude that our model consistently outperforms these variants with different gains. Our strategy does not require complicated operations and has fewer hyperparameters. Moreover, when removing the regularization term $\mathcal{L}_{\text{reg}}$, its performance decreases a little bit, which indicates the entropy constraints could help reduce noises in the predictions. We also investigate the transferability of different widely used graph neural network architectures, such as GCN [20], GraphSAGE [15], GAT [46] and GIN [53]. Their results are demonstrated in Table 5. Surprisingly, we observe that the simplest architecture GCN performs best among all the architectures, which is consistent with the results of no-adaptation methods.

**Embedding Visualization.** For a more intuitive grasp of the acquired target node representations, we project them into 2-D space via t-SNE [45]. The scatter plots of MDAN [63], GRADE [50], DECISION [1] and GraphATA are presented in Figure 5, where different colors indicate different different classes. Among them,

**Table 5: GraphATA results with different GNN architectures.**

| Architectures | A,D→C | C,D→A | A,C→D |
|---|---|---|---|
| GraphATA$_{\text{GCN}}$ | 82.33±0.75 | 73.17±0.52 | 78.45±0.87 |
| GraphATA$_{\text{SAGE}}$ | 79.41±0.27 | 69.47±0.86 | 73.83±0.86 |
| GraphATA$_{\text{GAT}}$ | 77.90±0.95 | 69.48±0.68 | 71.81±0.99 |
| GraphATA$_{\text{GIN}}$ | 77.06±0.33 | 65.77±0.25 | 72.10±1.29 |

two source-needed representatives MDAN and GRADE can not generate satisfactory visualizations, since there are no clear boundaries among these clusters, and nodes from different clusters are intertwined with each other. On the contrary, DECISION and our proposed GraphATA can cluster nodes together within the same classes, which validates the importance of distinguishing the contributions from different source domains. Furthermore, our GraphATA produces more compact clusters with clear boundaries.

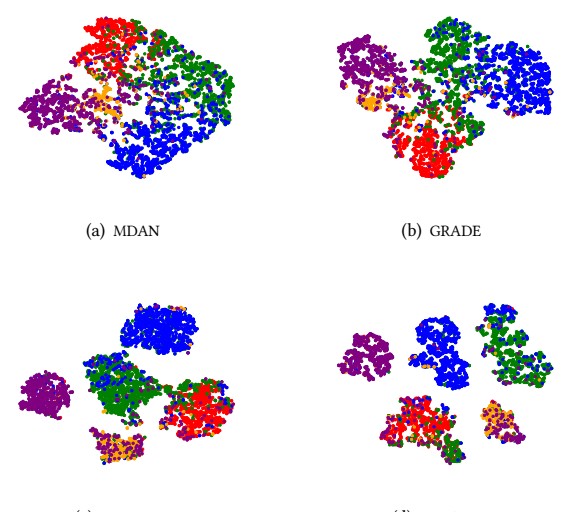

(a) MDAN

(b) GRADE

(c) DECISION

(d) GraphATA

**Figure 5: Node embedding visualizations in target graph, where colors correspond to different classes in citation networks (C,A→D).**

## 6 Conclusion

We investigate an important yet unexplored problem: unsupervised multi-source-free graph domain adaptation, which adapts multiple source pre-trained models without accessing the labeled source data. More specifically, we propose to perform node-centric instead of model-centric adaptation by parameterizing each node with its own graph convolutional matrix according to its local context information. The whole framework can be applied to various graph neural network architectures and downstream tasks including node as well as graph classification. We also illustrate that GraphATA can generalize to existing model-centric and layer-centric methods. Comprehensive experiments are conducted to show the effectiveness of our proposed GraphATA. For future work, it would be interesting to extend our model to more complex adaptation tasks like open-set graph domain adaptation and graph domain out-of-distribution, etc.

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

## A  Proof for Equation (7)

To summarize, sparsemax(·) considers the euclidean projection of the input vector $\boldsymbol{\alpha}$ onto the probability simplex, which can be defined as the following optimization problem:

$$\arg\min_{\boldsymbol{x}\in\Delta^{m-1}} \|\boldsymbol{x}-\boldsymbol{\alpha}\|^2, \; s.t., \; \mathbf{1}^\top \boldsymbol{x}=1, \; \boldsymbol{x}\geq \mathbf{0}. \tag{13}$$

Then, the Lagrangian of the optimization problem in Eq. (13) is:

$$\mathcal{L}(\boldsymbol{x},\boldsymbol{\mu},\omega) = \frac{1}{2}\|\boldsymbol{x}-\boldsymbol{\alpha}\|^2 - \boldsymbol{\mu}^\top \boldsymbol{x} + \omega(\mathbf{1}^\top \boldsymbol{x}-1). \tag{14}$$

The optimal $(\boldsymbol{x}^*,\boldsymbol{\mu}^*,\omega^*)$ must satisfy the following Karush–Kuhn–Tucker conditions:

$$\boldsymbol{x}^* - \boldsymbol{\alpha} - \boldsymbol{\mu}^* + \omega^*\mathbf{1} = 0, \tag{15}$$

$$\mathbf{1}^\top \boldsymbol{x}^* = 1, \; \boldsymbol{x}^* \geq \mathbf{0}, \; \boldsymbol{\mu}^* \geq \mathbf{0}, \tag{16}$$

$$x_i^* \mu_i^* = 0, \; \forall i \in \{1,\cdots,m\}. \tag{17}$$

If for $\forall i \in \{1,\cdots,m\}$, we have $x_i^* > 0$, then from Eq. (17) we must satisfy $\mu_i^* = 0$. Thus, from Eq. (15), we can get $x_i^* = \alpha_i - \omega^*$. Let $S(\boldsymbol{\alpha}) = \{j \in \{1,\cdots,m\} | x_j^* > 0\}$. From Eq. (16), we obtain $\sum_{j\in S(\boldsymbol{\alpha})}(\alpha_j - \omega^*) = 1$, which yields $\omega^* = \tau(\boldsymbol{\alpha})$ in Eq. (7). Again, from Eq. (17), we have that $\mu_i^* > 0$ implies $x_i^* = 0$, which from Eq. (15) implies $\mu_i^* = \omega^* - \alpha_i^* \geq 0$, i.e., $\alpha_i^* \leq \omega^*$ for $i \notin S(\boldsymbol{\alpha})$. That's to say, if the element $\alpha_i$ less than threshold $\omega^*$, then $\mu_i^*$ will larger than 0, and output $x_i$ must be reset to 0 to satisfy Eq. (17). Thus, we can generate the sparse values and have the property of sum-to-one.

## B  Datasets Details

In this section, we present the detailed information for experimental datasets including data processing and dataset splits. Specifically, we employ three types of graphs for the node classification task, which involve synthetic, social and citation networks. For synthetic datasets, contextual stochastic block models with different intra-class probability $p$ and inter-class probability $q$ are utilized to synthesize varying degrees of conditional structural shifts. Particularly, we fix intra-class probability $p = 0.04$ and vary inter-class probability $q$ from $\{0.012, 0.014, 0.016, 0.018\}$ to generate C1, C2, C3 and C4. As for node attributes, we construct a multivariate normal distribution for each class, where the mean vectors are set as $\{-2.0, -2/3, 2/3, 2\}$ with 128-dimension for each class and the covariance matrix is fixed as identity matrix. Then, each class's node attributes are sampled from those multivariate normal distributions. For Twitch datasets, we use their default splits, and different sub-datasets are constructed from distinct regions, which encompasses domain level distribution shifts across different datasets. For Citation dataset, the graphs are extracted from different platforms and periods, thus it contains both domain level and temporal level distribution shifts. For graph classification task, we employ three TUdatasets: Proteins, Mutagenicity, and Frankenstein, partitioning each dataset into four equally sized disjoint groups based on density shifts. The detailed information is presented in Table 1.

**Ethical Use of Data and Informed Consent.** All of our datasets are synthetic or publicly available, and do not involve human participants and subjects.

## C  Implementation Details

### C.1  Running Environment

Our experiments are conducted on a Linux server with 2 AMD EPYC 7543 CPU@2.80GHz, 512G RAM and one NVIDIA A100-SXM4-80GB GPU. The proposed model is implemented with Pytorch 1.13.1 in Python 3.8 using Pytorch Geometric 2.4.0.

### C.2  Compared Baselines

We present the detailed running configures for all the baselines as follows. Since some of the methods are initially devised for i.i.d image datasets, we modify their backbone to suit non-iid graph-structured data. The node representation dimension is set as 128 for node classification task and 64 for graph classification task, respectively.

*C.2.1  Source-Needed.*

- **MDAN.**[3] We use the source code released by the authors and replace its backbone with GCN. MDAN utilizes $K$ domain classifiers to implicitly align source and target representations via adversarial training. We use MDAN's Soft-Max version by setting the mode as dynamic and $\mu, \gamma$ are set as 0.01 and 10.0, respectively.
- **M³SDA.**[4] The codes are from the author's github page and we use GCN as the backbone. M³SDA employs moment matching to explicitly match source and target representations. We adopt 5-order moment matching in the experiments.
- **UDAGCN**[5], **GRADE**[6] and **SpecReg**[7] are three baselines designed for single source need graph domain adaptation. To adapt them to the scenario of multiple sources, we consolidate all the source domains into a single unified source domain. For UDAGCN, when generating PPMI matrix, we set the random walk length as 5 and each node is repeated 40 times. For SpecReg, the gamma smooth is set as 0.01.

*C.2.2 No-adaptation GNNs.*

- **GCN, SAGE, GAT, GIN.**[8] We use the implementations provided by Pytorch Geometric. The number of layers $L$ is set as 2. For GAT, we set number of attention heads as 1 and concat is initialized with False. For GIN, we utilize one MLP layer as the injective multiset function and the learning parameter $\epsilon$ is set as True. Finally, ReLU is used as the activation function and we fix the dropout ratio as 0.1 across different models.

*C.2.3 Single-source-free.*

- **SHOT**[9], **BNM**[10], **ATDOC**[11], **NRC**[12], **JMDS**[13] and **TPDS**[14] are six methods devised for single-source-free domain adaptation in the field of computer vision, thus we replace their backbones with GCN. In the context of multi-source-free domain adaptation, we normalize the soft predictions from multiple sources by taking their average. For ATDOC, the nearest centroid version is used. For NRC, we fix both neighbor size and neighbor of neighbor size as 3.
- **GTRANS**[15], **SOGA**[16] and **GraphCTA**[17] are three single-source-free graph domain adaptation methods. Similar to other baselines in this group, we average their predictions from multiple sources. For GTrans, the loops of adjacent matrix and node features are set as 1 and 4, respectively. For SOGA, the number of negative and positive samples are set as 5 and 2 in the NCE loss function.

---

[3] https://github.com/hanzhaoml/MDAN
[4] https://github.com/VisionLearningGroup/M3SDA
[5] https://github.com/GRAND-Lab/UDAGCN
[6] https://github.com/jwu4sml/GRADE
[7] https://github.com/Shen-Lab/GDA-SpecReg
[8] https://pytorch-geometric.readthedocs.io/en/latest/modules/nn.html
[9] https://github.com/tim-learn/SHOT
[10] https://github.com/cuishuhao/BNM
[11] https://github.com/tim-learn/ATDOC
[12] https://github.com/Albert0147/NRC_SFDA
[13] https://github.com/Jhyun17/CoWA-JMDS
[14] https://github.com/tntek/TPDS
[15] https://github.com/ChandlerBang/GTrans
[16] https://github.com/HaitaoMao/SOGA
[17] https://github.com/cszhangzhen/GraphCTA

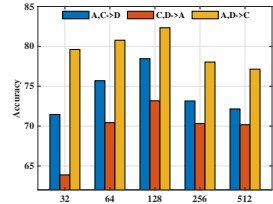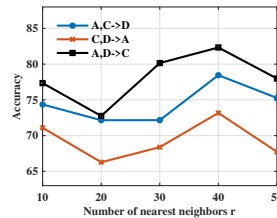

**Figure 6: Hyper-parameter sensitivity analysis.**

*C.2.4 Multi-source-free.*

- **DECISION**[18], **CAiDA**[19] and **MSFDA**[20] are three recent multi-source-free domain adaptation models that are particularly designed for i.i.d image classification task. We replace their backbone with GCN as other baselines. For CAiDA, we use cosine similarity as the distance function to construct confident anchors. For MSFDA, we reuse their default hyperparameters.

# D Time and Space Complexity Comparisons

We compare our proposed model with DECISION [1], a representative model-centric multi-source-free domain adaptation method. Specifically, given a graph with $n$ nodes and $e$ edges, the node representation dimension is set as $d$ and GNN has $L$ layers. $K$ is the number of categories. Then, if we have $m$ source models from different domains, the feature encoder has the time complexity of $O(Lmnd^2 + Lmed)$. The time complexity of calculating cluster centroids for $m$ models is $O(mnd)$. Computing the pseudo-label of each sample by assigning it to its nearest cluster centroid has a time complexity of $O(mnKd)$ for $m$ models. Thus, the overall time complexity is $O(Lmnd^2 + Lmed + mnd + mnKd)$. Meanwhile, to store $m$ models' predictions and node representations, it requires a space complexity of $O(mnd + mnK)$. The time complexity of our proposed GraphATA has been discussed in the end of Section 4. For ease of comparison, we present the time and space complexity of DECISION and GraphATA in Table 6. Among them, $|G|$ refers to the model parameters of the Graph Neural Network (GNN), while $|S|$ represents the additional parameters of our model, such as $W_g^l$ and $a^l$. Our model utilizes aggregated weight matrices to perform graph convolution once, while DECISION performs graph convolution $m$ times and then aggregates their predictions. Similarly, GraphATA maintains one node representation bank and one prediction bank, while DECISION stores them $m$ times. Thus, GraphATA has lower time and space complexity compared with DECISION. It's worth noting that other baselines like CAiDA [10] and MSFDA [39] exhibit higher time complexity compared to DECISION, because they utilize more complicated strategies to generate pseudo-labels. In summary, our model demonstrates the lowest time and space complexity among the compared multi-source-free methods.

**Table 6: Comparisons of time and space complexity.**

|  | DECISION | GraphATA |
|---|---|---|
| Time | $O(Lmnd^2 + Lmed + mnd + mnKd)$ | $O(Lnd^2 + Led + mnd + nmlog(m) + dnlog(n))$ |
| Space | $O(m|G| + mnd + mnK)$ | $O(m|G| + nd + nK + |S|)$ |

---

**Algorithm 1:** Pseudocode of GraphATA

**Input**   : $m$ source pretrained graph neural network models $\{\Phi_1, \cdots, \Phi_m\}$ and an unlabelled target graph $\mathcal{G} = (\mathbf{A}, \mathbf{X})$ or a set of unlabeled graphs $\{\mathcal{G}_1, \cdots, \mathcal{G}_n\}$

**Output**: model $\Phi_t$ for target graph classification

1  Load model weights from source pretrained models $\{\mathbf{W}_1^l, \cdots, \mathbf{W}_m^l\}_{l=1,\cdots,L}$

2  **while** *not converged or not reached the maximum epochs* **do**

3     **for** *layer* $l \leftarrow 1$ **to** $L$ **do**

4         Compute personalized graph convolutional matrix for each node $v$ via Eq. (4)

5         Perform graph convolution at layer $l$

6     Compute node representations $\mathbf{Z} \in \mathbb{R}^{n \times h}$ and predictions $\mathbf{P} \in \mathbb{R}^{n \times C}$ with $\Phi_t$

7     Calculate loss $\mathcal{L}$ according to Eq. (12) and update model $\Phi_t$'s parameters

8     Update memory banks $\mathcal{R}$ and $\mathcal{P}$ via a momentum manner

9  Compute predictions $\mathbf{Y} = \Phi_t(\mathbf{A}, \mathbf{X})$ with optimized model $\Phi_t$

---

**Table 7: GraphATA results with different local contexts.**

| Context | A,C→D | C,D→A | A,D→C |
|---|---|---|---|
| $\mathbf{c}_{max}$ | 75.67±1.21 | 71.66±1.15 | 78.09±2.01 |
| $\mathbf{c}_{min}$ | 74.63±2.16 | 70.40±1.07 | 79.93±1.18 |
| $\mathbf{c}_{sum}$ | 72.50±1.44 | 70.01±2.75 | 80.10±1.02 |
| $\mathbf{c}_{mean}$ | 78.45±0.87 | 73.17±0.52 | 82.33±0.75 |

## E   More Ablation Studies and Experiments

**Hyperparameter Analyses.** We present the sensitivities of node representation dimension and number of nearest neighbors in Figure 6. Particularly, node dimension $d = 128$ and $r = 40$, our model could always obtain the satisfied performance. As we can see, accuracy improves consistently as the node representation dimension increases from 32 to 128, then slightly drops at 256. For number of nearest neighbors, the accuracy remains relatively stable, fluctuating between 75% and 80%, with a slight dip at 20 neighbors, then recovering toward 40 neighbors. This is because too few nearest neighbors could not provide sufficient supervision information, while too many nearest neighbors might introduce noises into the generation process. While there are fluctuations, they remain within a reasonable range, as our sensitivity analysis covering a wide range of nearest neighbor values.

**The Effect of Different Aggregation Strategies for Local Contexts.** The mean operation is chosen for its simplicity in aggregating information from neighboring nodes. It provides a straightforward way to capture the average characteristics of the neighborhood, which is widely adopted in many graph-based learning models. While it is true that the mean operation can sometimes

---

18https://github.com/driptaRC/DECISION
19https://github.com/Learning-group123/CAiDA
20https://github.com/maohaos2/MSFDA

**Table 8: GraphATA results on ogbn-arxiv datasets.**

| Methods | 2016-2018 | 2018-2020 |
|---|---|---|
| GCN [20] | 55.57±0.09 | 53.03±0.16 |
| SOGA [33] | 58.10±0.23 | 52.28±0.12 |
| DECISION [1] | 59.53±0.17 | 57.55±0.34 |
| CAiDA [10] | 58.42±0.14 | 56.19±0.38 |
| MSFDA [39] | 61.78±0.87 | 59.91±0.20 |
| GraphATA | **63.55±0.94** | **60.85±0.13** |

**Table 9: GraphATA results on TRIANGLE datasets.**

| Methods | T1,T2,T4→T3 | T1,T2,T3→T4 |
|---|---|---|
| DECISION [1] | 31.90±0.61 | 19.40±0.23 |
| CAiDA [10] | 40.23±0.39 | 17.05±0.77 |
| MSFDA [39] | 39.46±0.59 | 19.04±1.46 |
| GraphATA | **43.08±0.77** | **22.49±0.34** |

result in similar **c** values for nodes with different types of neighbors, our ablation studies demonstrate that our model maintains high performance when compared to other aggregation strategies such as max, min and sum, as shown in the Table 7. These results highlight the robustness of our mean aggregation approach despite its simplicity.

**Additonal Experimental Results.** We conduct experiments on a larger citation dataset **ogbn-arxiv** for node classification task, which contains 169, 343 nodes and 1, 166, 243 edges from 40 classes. The dataset is chronologically divided into five groups according to the publication years of the papers. *We construct three source graphs encompassing papers published before 2011, during the periods 2011-2014 and 2014-2016, while the two target graphs are derived from the periods 2016-2018 and 2018-2020.* The results are presented in Table 8. As demonstrated in the table, our proposed GraphATA consistently

**Table 10: Average node and graph classification performance in terms of accuracy with standard deviation (%).**

| Models | C1,C2,C4 → C3 | C1,C3,C4 → C2 | C2,C3,C4 → C1 | P1,P2,P4 → P3 | P1,P3,P4 → P2 | P2,P3,P4 →P1 |
|---|---|---|---|---|---|---|
| MDAN | 89.65±0.15 | 91.88±0.31 | 91.33±0.28 | 67.45±0.17 | **77.66±0.18** | 54.10±0.25 |
| GCN | 86.14±0.16 | 87.75±0.08 | 76.46±0.12 | 63.44±1.07 | 72.30±0.71 | 55.57±1.34 |
| DECISION | 91.05±0.74 | 92.15±0.81 | 92.05±0.56 | 65.23±1.43 | 70.32±0.89 | 71.04±0.93 |
| CAiDA | 90.31±0.19 | 91.37±0.20 | 91.61±0.56 | 64.69±0.18 | 71.40±0.53 | 67.62±0.67 |
| MSFDA | 92.87±0.49 | 92.75±0.28 | 93.31±0.14 | 67.02±1.17 | 72.48±0.89 | 56.51±0.53 |
| GraphATA | **93.49±0.71** | **93.85±0.42** | **94.62±0.84** | **68.48±0.16** | 74.19±0.13 | **72.47±0.85** |
| | M1,M2,M4 → M3 | M1,M3,M4 → M2 | M2,M3,M4 → M1 | F1,F2,F4 → F3 | F1,F3,F4 → F2 | F2,F3,F4 →F1 |
| MDAN | 76.91±0.78 | **82.05±1.15** | 70.89±0.23 | 50.18±0.46 | 48.48±0.23 | 58.90±0.32 |
| GCN | 78.24±0.92 | 77.30±1.42 | 71.54±0.96 | 55.29±0.69 | 52.76±0.36 | 56.13±1.06 |
| DECISION | 68.89±1.79 | 64.21±1.84 | 58.16±0.51 | 54.33±1.24 | 51.75±0.46 | 53.82±0.69 |
| CAiDA | 66.35±1.65 | 69.28±1.39 | 63.05±0.50 | 53.92±1.47 | 48.00±0.55 | 56.54±2.02 |
| MSFDA | 76.63±0.50 | 74.21±1.89 | 65.21±1.64 | 51.15±0.89 | 48.52±0.96 | 53.69±0.36 |
| GraphATA | **80.62±0.33** | 78.54±0.17 | **72.67±0.32** | **57.46±0.64** | **54.78±0.83** | **60.45±0.22** |
| | R,P,F,EN, ES→ DE | R,P,F,EN, DE→ ES | R,P,F,DE, ES→ EN | P,F,DE,EN, ES→ R | R,F,DE,EN, ES→ P | R,P,DE,EN, ES→F |
| MDAN | 58.60±0.23 | 70.83±0.10 | 50.46±0.58 | 73.94±0.12 | 65.74±0.31 | 63.20±0.15 |
| GCN | 47.08±0.33 | 71.32±0.20 | 48.85±0.21 | 69.61±0.30 | 68.04±0.50 | 64.90±0.63 |
| DECISION | 60.90±0.98 | 61.86±0.65 | 58.36±0.54 | 66.60±0.86 | 67.01±0.15 | 77.26±0.32 |
| CAiDA | 53.79±0.84 | 68.13±0.70 | 58.49±0.56 | 69.63±0.14 | 68.34±0.27 | 72.60±0.64 |
| MSFDA | 59.40±1.93 | 70.74±0.75 | 54.68±1.00 | 75.41±0.68 | 65.08±0.54 | 78.06±1.80 |
| GraphATA | **62.49±0.12** | **72.55±0.38** | **59.57±0.61** | **77.50±0.15** | **70.58±0.51** | **80.21±0.17** |

exhibits effective performance across various adaptation scenarios within this large-scale citation dataset.

We also choose the **TRIANGLE** dataset, a large-scale dataset from TUDataset for graph classification task, which consists of 45000 graphs across 10 classes. Then, we partition it into four equally size disjoint groups based on density shift (i.e., T1,T2,T3,T4) and compare our model with 3 recent multi-source-free baselines. The results are demonstrated in Table 9. As shown in the table, our proposed model consistently demonstrates strong performance across a range of adaptation scenarios within this large-scale dataset.

Finally, Table 10 presents all the adaptation results on social datasets and TUDatasets, which further verify the effectiveness of our proposed GraphATA under different settings.

**GPU Consumption and Training Time per Iteration.** We measured the GPU consumption and training time per epoch for our proposed GraphATA and compared it with representative methods as follows. GCN is trained independently on each labeled source domain and then directly evaluated on the target domain without any adaptation process. Therefore, we do not report the GPU consumption and training time for GCN. We also do not report the results of single-source-free models, as they require adapting each of the $m$ source models to the target domain individually. Instead, we focus on reporting the adaptation time for source-needed as well as multi-source-free models to provide a meaningful comparison

**Table 11: GPU memory and time in C,D→A.**

| Methods | GPU | Time |
|---|---|---|
| MDAN [63] | 12800MB | 0.1075s |
| M3SDA [37] | 17688MB | 0.1198s |
| DECISION [1] | 6920MB | 1.5050s |
| CAiDA [10] | 6880MB | 3.6579s |
| MSFDA [39] | 10080MB | 9.1350s |
| GraphATA | 5930MB | 0.0893s |

**Table 12: GPU memory and time in P1,P2,P3→P4.**

| Methods | GPU | Time |
|---|---|---|
| MDAN [63] | 756MB | 0.0401s |
| M3SDA [37] | 3898MB | 0.1144s |
| DECISION [1] | 674MB | 0.0428s |
| CAiDA [10] | 668MB | 0.0468s |
| MSFDA [39] | 684MB | 9.3613s |
| GraphATA | 596MB | 0.0380s |

of their performance in adapting to the target domain. The results are shown in Table 11 and Table 12, which is consistent with our analyses in previous sections.

