# OpenReview forum: "Aggregate to Adapt: Node-Centric Aggregation for Multi-Source-Free Graph Domain Adaptation"
_ACM.org/TheWebConf/2025/Conference — WWW 2025 Poster_

### Official Review · Reviewer_3XB8 · 2024-11-07

**Novelty:** 4
**Technical Quality:** 5

**Review:**

The paper presents an approach to unsupervised graph domain adaptation (UGDA), addressing the challenges posed by domain discrepancies in graph data, particularly when transferring knowledge from multiple source domains to an unlabeled target domain. The proposed method emphasizes the importance of aggregating information from various sources while considering the unique contributions of each domain.
Weakness:
1. the node-centric approach will increase the computation complexity, which could be impractical for real application.
2. some recent works also follow node-centric approach, though they are not focused on the same task. For example, The Snowflake Hypothesis: Training and Powering GNN with One Node One Receptive Field.
3. the authors point out the drawbacks of model-centric approaches, thus, comparisons with model-centric approaches are desired.

**Questions:**

1.Why the used datasets have a small size？

**Reviewer Confidence:**

3: The reviewer is confident but not certain that the evaluation is correct

**Scope:**

4: The work is relevant to the Web and to the track, and is of broad interest to the community

---

### Official Review · Reviewer_CZML · 2024-11-21

**Novelty:** 4
**Technical Quality:** 4

**Review:**

The paper presents a novel approach for multi-source-free unsupervised graph domain adaptation (MSFDA) with a node-centric framework. The methodology is technically sound, with a clear derivation of the proposed GraphATA model, robust theoretical underpinnings, and extensive experiments validating its performance. The incorporation of sparse attention mechanisms and personalized convolutional matrices effectively addresses limitations in prior model- or layer-centric approaches.

Pros
1. Addresses a Practical Problem: The problem of MSFDA without labeled source data is critical for real-world applications involving privacy constraints.
2. Methodological Integration: Combines sparsemax attention with graph convolution, achieving fine-grained node-centric adaptation.
3. Comprehensive Experiments: Evaluates performance across diverse datasets with extensive ablation studies.

Cons
1. Lack of Theoretical Innovation: While the proposed node-centric adaptation framework is interesting, it primarily builds upon existing techniques such as sparsemax and graph convolutional networks (GCNs). The work does not introduce fundamentally new theoretical insights or methodologies.
2. High Computational Complexity: The node-centric adaptation requires dynamically generating convolutional matrices for each node, which could lead to significant computational and storage overhead, especially for graphs with millions of nodes and edges. The paper lacks empirical results to demonstrate the method’s scalability on large-scale graphs.
3. Incomplete Codebase: The provided codebase lacks a requirements.txt file, which significantly affects reproducibility.

**Questions:**

1. Node-Centric Attention: Could you clarify how the sparsemax function improves over softmax in terms of computational efficiency for large graphs?
2. Scalability: Have you tested GraphATA on significantly larger graphs, and if so, what were the computational challenges?

**Ethics Review Description:**

NULL

**Ethics Review Flag:**

Yes

**Reviewer Confidence:**

4: The reviewer is certain that the evaluation is correct and very familiar with the relevant literature

**Scope:**

4: The work is relevant to the Web and to the track, and is of broad interest to the community

---

### Official Review · Reviewer_aNbD · 2024-11-29

**Novelty:** 4
**Technical Quality:** 4

**Review:**

**Quality**: The methodologies proposed in the paper are well-reasoned and technically feasible. The experimental setup is thorough and sufficiently detailed. However, the algorithm design, which focuses on model weights, presents certain limitations and implementation challenges, restricting its scalability to new models or models with a large number of parameters.

**Clarity**: In the related work section, the authors are encouraged to provide a clearer differentiation and connection between MSFDA, MSDA, and single-source-free domain adaptation. A comprehensive review of the developmental trajectory of this field would enhance clarity and contextual understanding, rather than selectively highlighting specific shortcomings or techniques when discussing limitations.

**Originality**: The paper exhibits a notable degree of originality by converting the contribution ratios of classification predictions from multi-source pre-trained models into model weight ratios. This approach effectively integrates source domain knowledge from multi-source pre-trained models into the prediction process, while quantifying the respective contributions of each source.

**Significance**: The work’s significance is moderate, as it presents certain limitations. The decomposition and unification of multi-source pre-trained model weights, along with the requirement for algorithmic architectures to align closely with the structure of pre-trained models, constrain its broader applicability and impact.

**Questions:**

- The paper does not address unsupervised graph representation algorithms. It is worth noting that most unsupervised graph representation learning approaches inherently possess certain domain adaptation properties. In my view, some studies on feature invariance based on unsupervised graphs essentially fall under the scope of MSFDA. If a simple augmentation of the dataset were used to align with the current experimental setup, would the method still demonstrate a significant advantage in prediction accuracy?

- The algorithm design exhibits certain limitations, as it is restricted to scenarios where multi-source pre-trained models are used in conjunction with GNN networks. Such constraints are uncommon in MSFDA experimental setups. If the multi-source pre-trained models incorporate diverse network architectures, or even models with a large number of parameters, how would the algorithm design proposed in the paper adapt to these scenarios?

**Reviewer Confidence:**

3: The reviewer is confident but not certain that the evaluation is correct

**Scope:**

3: The work is somewhat relevant to the Web and to the track, and is of narrow interest to a sub-community

---

### Official Review · Reviewer_MSUh · 2024-11-30

**Novelty:** 5
**Technical Quality:** 5

**Review:**

This paper addresses unsupervised multi-source-free graph domain adaptation, introducing a novel node-centric adaptation approach. By parameterizing each node with a personalized graph convolutional matrix based on its local context, the framework achieves fine-grained alignment. The method is versatile, applicable to various graph neural network architectures and downstream tasks.

Pros:
New Setting Exploration: The paper ventures into a relatively unexplored area—multi-source-free unsupervised graph domain adaptation—offering fresh insights into this challenging problem.
Innovative Node-Level Parameterization: Personalized graph convolutional matrices represent a significant departure from traditional model-level adaptations, enabling more precise alignment to target graphs.
Strong Empirical Performance: The proposed model consistently outperforms state-of-the-art baselines across multiple datasets and tasks. Results are bolstered by robust ablation studies and hyperparameter analyses, demonstrating the framework's effectiveness.

Cons:
1, The assumption that all source models provide meaningful knowledge for every node may be overly optimistic. In real-world scenarios, unaligned graph structures often hinder uniform adaptation.

2,  While the methodology is innovative, a more detailed discussion of why existing frameworks fail to adapt to dynamic neighborhood information would better contextualize the contributions.

3,  The model’s complexity scales with the number of source domains and layers, potentially limiting its practicality for large-scale graphs or highly fragmented sources.

**Questions:**

See Cons.

**Reviewer Confidence:**

3: The reviewer is confident but not certain that the evaluation is correct

**Scope:**

3: The work is somewhat relevant to the Web and to the track, and is of narrow interest to a sub-community

---

### Official Review · Reviewer_HnQ2 · 2024-12-02

**Novelty:** 5
**Technical Quality:** 5

**Review:**

The authors study a new task of Multi-Source-Free Graph Domain Adaptation and propose a node-centric adaptation framework to aggregate models from multiple source domains. Experimental results show that the proposed method achieves better performance than other baselines.
Pros
1.	The task of multi-source-free unsupervised graph domain adaptation is notable and valuable.
2.	The proposed method is clearly described and outperforms other baselines.
3.	The authors provide comprehensive experimental settings, results analysis and theoretical proofs, enhancing the quality of the method.
Cons
1.	The aggregation of graph convolutional matrices in Eq. (4) imposes restrictive constraints on the network structures across different source GNN models, which limits the generalizability of the proposed method. Specifically, it requires that different source GNN models maintain identical layer depth $l$ and identical dimension of the convolutional matrix $W^l_i$.
2.	To implement the proposed method, the authors impose identical node representation dimensions across GNN models from different sources in the experiment (128 for node classification task and 64 for graph classification task). The uniform node representation dimension may not necessarily ensure that the models used for aggregation represent the optimal performance for each individual source graph. If the most suitable model architecture could be identified and optimally trained for each source graph, other architecture-independent baselines may demonstrate better performance.

**Questions:**

How would the proposed method handle scenarios with graph convolutional matrices of different dimensions or non-uniform shapes across diverse source models?

**Reviewer Confidence:**

3: The reviewer is confident but not certain that the evaluation is correct

**Scope:**

4: The work is relevant to the Web and to the track, and is of broad interest to the community